# Chasing the Apomictic Factors in the *Ranunculus auricomus* Complex: Exploring Gene Expression Patterns in Microdissected Sexual and Apomictic Ovules

**DOI:** 10.3390/genes11070728

**Published:** 2020-06-30

**Authors:** Marco Pellino, Diego Hojsgaard, Elvira Hörandl, Timothy F. Sharbel

**Affiliations:** 1Leibniz Institute of Plant Genetics and Crop Plant Research (IPK), D-06466 Gatersleben, Germany; marco.pellino@gifs.ca (M.P.); tim.sharbel@usask.ca (T.F.S.); 2Seed Developmental Biology Program, Global Institute for Food Security (GIFS), University of Saskatchewan 105 Administration Place, Saskatoon, SK S7N 5A2, Canada; 3Department of Systematics, Biodiversity and Evolution of Plants, Albrecht-von-Haller Institute for Plant Sciences, Georg-August-University of Göttingen, Untere Karspüle 2, D-37073-1 Göttingen, Germany; dhojsga@gwdg.de

**Keywords:** apomixis, differentially expressed genes, hybridization, microarrays, polyploidy, *Ranunculus*, sexuality

## Abstract

Apomixis, the asexual reproduction via seeds, is associated to polyploidy and hybridization. To identify possible signatures of apomixis, and possible candidate genes underlying the shift from sex to apomixis, microarray-based gene expression patterns of live microdissected ovules at four different developmental stages were compared between apomictic and sexual individuals of the *Ranunculus auricomus* complex. Following predictions from previous work on mechanisms underlying apomixis penetrance and expressivity in the genus, gene expression patterns were classified into three categories based on their relative expression in apomicts compared to their sexual parental ancestors. We found evidence of misregulation and differential gene expression between apomicts and sexuals, with the highest number of differences detected during meiosis progression and emergence of aposporous initial (AI) cells, a key developmental stage in the ovule of apomicts where a decision between divergent reproductive pathways takes place. While most of the differentially expressed genes (DEGs) could not be annotated, gene expression was classified into transgressive, parent of origin and ploidy effects. Genes related to gametogenesis and meiosis demonstrated patterns reflective of transgressive and genome dosage effects, which support the hypothesis of a dominant factor controlling apomixis in *Ranunculus* and modulated by secondary modifiers. Three genes with probable functions in sporogenesis and gametogenesis development are identified and characterized for future studies.

## 1. Introduction

The evolution of asexuality from sexual progenitors has occurred repeatedly and independently in a wide range of plants and animals [1,2,3]. The evolutionary consequences of such a drastic functional trait change have been the subject of many studies and have shed light onto mechanisms of species dispersal and adaptation relevant to the evolutionary success of asexuality [4,5]. It has been suggested that asexual reproduction is advantageous because it reduces to a half the costs associated with sex [6] and because of a greater adaptability to stable environments [7]. In addition, asexuality would be beneficial for fixing advantageous mutations and genotypes highly adapted to the surrounding environment [8,9]. Despite these expected advantages, limitations arise from the lack of meiotic recombination and exploitation of recombinational DNA repair mechanisms [10]. The absence of meiosis has also drastic consequences on purging deleterious mutations [11]. Asexuals are expected to accumulate deleterious mutations in a non-reversible ratchet-like fashion [12], a process which accelerates in small populations and can drive a lineage to extinction due to mutational meltdown [13]. In contrast, sexual recombination allows the purging of deleterious mutations [12,14] and provides statistical chances for advantageous mutations from distinct individuals to recombine together in a single offspring [8]. Moreover, sexuality is advantageous in changing environments whereby the exploitation of genetic variability through meiotic and gametic recombination is central for adaptation in natural populations [15,16].

Asexuality in plants can happen by different forms of vegetative propagation (like runners, bulbils, tubers, etc.), or by apomixis (clonal seed formation [17]). Apomixis has been described in more than 400 species scattered throughout 293 angiosperms genera [18,19]. Apomixis is sporophytic when an embryo develops directly from a somatic cell of the ovary, or gametophytic when the embryo develops through the formation of an unreduced female gametophyte [17]. Two main types of gametophytic apomixis have been characterized following the formation of an unreduced embryo sac from a nucellar cell (apospory), or from the megaspore mother cell (MMC) via an altered meiosis (diplospory). In all cases of gametophytic apomixis, three developmental steps must sequentially occur to produce a functional clonal seed: (1) the development of a meiotically unreduced egg cell (apomeiosis), (2) the development of the seed’s embryo without fertilization (parthenogenesis), and (3) the production of a functional endosperm with (pseudogamy) or without (autonomous) fertilization [20].

The molecular control of apomixis is still uncertain. Different hypotheses have been proposed to explain observations and experimental data from diverse apomictic plant species. Perhaps the two most accepted hypothesis are the idea of apomixis being a consequence of changes to developmental timing of reproduction, and the one considering apomixis as a mutation-based phenomenon [21,22,23].

In the first case, since polyploidy and hybridization are common traits of almost all apomictic plants [20], both have been hypothesized to be triggers for the reproductive switch from sexuality to apomixis. In this theoretical frame, it is proposed that global gene regulatory changes that follow polyploidization and hybridization in general, and specifically de-regulation of the standard sexual reproductive program lead to apomixis in plants [18], and can arise via chromosomal [24,25,26], genomic [27,28], and transcriptomic [26,29] changes in gene regulatory networks. As support for this hypothesis, studies demonstrated heterochronic gene expression (i.e., deregulation) between sexual and apomictic reproductive tissues in *Tripsacum* spp. [30] and *Boechera* spp. [31].

In the second case, apomixis in plants has been often identified with one or two dominant factors segregating according to Mendelian proportions, and which are altered via epistasis, genetic modifiers, polyploidy, etc. [32]. In several studies using different apomictic species of the *Ranunculus auricomus* complex throughout four generations of crossings and backcrossings, Nogler [33,34,35] established that apomixis was controlled by a dominant factor A^−^ with a recessive lethal behaviour. One of the main consequences of such conclusion was that A^−^ could not be transmitted though haploid gametes, and thus nicely provided a causality for the intriguing rarity of apomixis in natural diploid plants. In his studies, Nogler [35,36] further concluded that the frequency (i.e., expressivity) of apomixis in individual plant genotypes was dosage dependent and relative to the wild type A^+^. More recently, part of Nogler’s results were contradicted by crossing experiments on other, more distantly related species within the *R. auricomus* complex, in which experimental diploid hybrids displayed functional apomixis in low frequencies and inheritance of apospory in diploids [37]. In this study, the expression of apospory was dosage-dependent and higher in plants with two aposporous parents compared to those with just one aposporous parent [37]. The apospory-specific factor appears to be not necessarily lethal, but allelic dosage effects were confirmed.

The *Ranunculus auricomus* complex is becoming a model for studying genomic differences between sexual and apomictic taxa. Hundreds of apomictic species are derived from a handful of sexual progenitor species [38,39]. The closely-related diploid sexual species *R. carpaticola*, and diploid and autotetraploid *R. cassubicifolius* extend from Switzerland to west Slovakia [39], while hexaploid apomicts are located in central Slovakia. Analyses based upon AFLPs and SSRs, combined with DNA sequence [39,40] and RNAseq SNPs [41], support a hybrid origin for hexaploid apomicts resulting from a cross between a putative 4x *R. cassubicifolius* and a 2x *R. carpaticola* sometime during the last glacial period (60,000–80,000 years ago). Apomictic *Ranunculus* species are aposporous (*Hieracium*-type), pseudogamous and express apomixis facultatively [36,37,42,43,44,45,46]. Aposporous initial (AI) cells arise from a somatic cell of the nucellus during the time in which the megaspore mother cell (MMC) starts meiosis [35,43,44], and competes for the formation of the female gametophyte within the ovule. Once the mature female gametophyte is formed, fertilization of the egg and central cells is required in meiotic female gametophytes while in apomictic ones only the central cell needs to be fertilized to form the endosperm [36,43,44].

Taking advantage of a previous study using SNPs and comparing sexual and asexual *Ranunculus* genotypes using Illumina RNAseq data [41], we developed a custom expression microarray and used it to transcriptionally profile microdissected ovules at four developmental stages from sexual diploid *R. carpaticola* and tetraploid *R. cassubicifolius* genotypes, and hexaploid apomictic *R. carpaticola* × *cassubicifolius* genotypes. Using SNPs and RNAseq data might be of help to distinguish the molecular cause for apomixis in *Ranunculus*, i.e., whether is due to heterochronic gene expression or dominant factors. Based on levels of gene expression changes and dysregulated genes, we would expect global changes in gene expression in the first case or rather a few changes on master genes in the second case. This, however, is at least challenging since one or a few dominant factors regulating apomixis would likely trigger a wide range of downstream changes in gene regulatory networks, blurring possibilities to segregate expression patterns into one of both hypotheses. Either way, our experimental approach and in silico analyses were designed to overcome difficulties due to the occurrence of differing ploidies in our target *Ranunculus* plants, exploiting multiple genotypes as biological replicates. The goals of this study were to (1) identify transcripts showing differential expression between apomictic and sexual ovules, (2) quantify heterochronic gene expression patterns during apomictic and sexual ovule developments, (3) partition transcripts into groups whose expression signatures reflect hybridity, parent of origin effects, or polyploidy, and (4) frame our findings according to previous and ongoing studies to understand the inheritance of apomixis in *Ranunculus* and identify evidence of molecular evolution between reproductively relevant sequences within the complex.

## 2. Materials and Methods

### 2.1. Custom Microarray Development

The RNAseq approach used to generate the data for microarray design was explained in detail in Pellino et al. [41]. In short, flowers were collected from two sexual and two apomictic genotypes of the *Ranunculus auricomus* complex within a two-week time window (Table 1). For each individual, total RNA was isolated from pooled flowers of five different sizes (two to seven mm in length, from an early stage when sepal primordia have enclosed the floral meristem, to the fully developed unopened flower stage) using the Qiagen RNeasy Plant Mini Kit (Qiagen GmbH, Hilden, Germany). After isolation, any DNA contamination was removed using Qiagen RNase-Free DNase, while contamination from the DNase enzyme, polysaccharides, and proteins were removed with a second purification step using the Qiagen RNeasy Mini Kit. The manufacturer’s instructions were followed in all purification steps.

To avoid over-representation of highly transcribed genes during subsequent sequencing steps, full length normalized cDNA libraries were produced, whereby each step of the normalization was performed and optimized following the procedures described by Vogel et al. [49] and Vogel and Wheat [50].

15 μL of normalized cDNA (200 ng/μL) was sent to Fasteris SA (Geneva, Switzerland) for RNAseq, where a dual-sequencing approach (54-mer single read (SE) and 108-mer pair-end read (PE)) was chosen in order to balance cost and efficiency of a de novo assembly (see Pellino et al. [41]). Both sequencing strategies were conducted using the HiSeq^TM^ 2000 sequencing system (Illumina Inc., San Diego, CA, USA). 

CLC Genomics Workbench (CLC bio version 4.9) was implemented for the sequencing assembly. At first, sequences were trimmed for vector contamination, length and quality score using CLC default values. As no reference genome for *Ranunculus* is available, de novo and iterated de novo approaches were used to assemble the data. For the de novo assembly, all libraries were pooled and assembled using CLC default parameters. In the iterated de novo assembly, each library was assembled individually, and the resultant contigs from each individual assembly re-assembled together, including all unassembled reads from each individual assembly. In both strategies the final assembly was trimmed for contigs shorter than 300 bp (see details in Pellino et al. [41]). The two approaches were evaluated based on the total number of matching reads (i.e reads that could be assembled into longer contigs) and N50 values, and the de novo assembly was selected for array design. Both contigs and singletons were forwarded to Roche NimbleGen Inc. (Madison, WI, USA) for design and manufacture. The NimbleGen selection strategy bioinformatically designed three different probes for each contig, and one to three probes for each singleton, followed by design of a 3 × 1.4 million-spot array. The final array therefore contained multiple technical replicates for each gene expressed during flower development in *Ranunculus*. All the microarrays data has been uploaded to Arrayexpress at EMBL-EBI (“Experiment E-MTAB-3316”).

### 2.2. Transcriptomal Profiling of Sexual and Apomictic Ovules

#### 2.2.1. Sample Selection, Ovule Microdissection, and RNA Extraction

Plants were grown from seedling to pre-flowering stages in outdoor plots at the Leibniz Institute of Plant Genetics and Crop Plant Research (IPK), and were then moved into a phytotron for flowering (day: 16 h, 21 °C; night: 8 h, 16 °C; humidity 70%; light intensity: 150 μmol/m^2^). Based on the cytological observations of Hojsgaard et al. [44], ovules at four developmental stages were collected at the pre-meiotic, meiotic (tetrad stage/aposporous initial), 2–4 nuclei embryo sac and mature embryo sac stages (I, II, III, IV; Appendix A) at standardized times (between 7:00 and 9:00 AM) over multiple days from both sexual and apomictic *Ranunculus* under a sterile laminar-flow hood with a stereoscopic microscope (100 Stemi; Carl Zeiss AG, Oberkochen, Germany). Using sterile scalpel and forceps to open the flower, carpels were collected and immediately immersed in a sterile 0.55 M mannitol solution and placed on ice.

In the second step, microdissection of each single carpel was conducted in a in a sterile laminar air-flow hood under an inverted microscope (Axiovert 200M; Carl Zeiss AG, Oberkochen, Germany) using hand-crafted sterile glass needles (self-made with a Narishige PC-10 puller). Individual ovules were collected using an Eppendorf Cell Tram Vario connected to a 150 μm inner diameter glass capillary, immersed in 100 μL RNA-stabilizing buffer (RNA later; Sigma-Aldrich, St. Luis, MO, USA), and immediately frozen in liquid nitrogen. 20 to 40 ovules for each genotype/stage were dissected and stored at −80 °C. RNA extraction was carried out using a PicoPure isolation kit (Thermo Fisher Scientific, Carlsbad, CA, USA) and quantification and quality were assessed with RNA Pico chips on an Agilent 2100 Bioanalyzer (Agilent Technologies, Santa Clara, CA, USA).

#### 2.2.2. cDNA Synthesis and Amplification

For each of the samples from the seven *Ranunculus* individuals (Table 1), cDNA synthesis and amplification were conducted using the Sigma TransPlex Complete WTA2 kit (Sigma-Aldrich) following the producer’s instructions. Amplified cDNA was purified using the GenEluteTM PCR Cleanup kit (Sigma-Aldrich) following the manufacturer’s protocol, and concentration and quality were measured with a NanoDropTM 1000 Spectrophotometer (Thermo Fisher Scientific, Wilmington, DE, USA).

#### 2.2.3. Microarray Hybridization and Data Processing

Twenty-eight 1.4 M probe custom microarrays were outsourced to Roche NimbleGen for design and manufacture, each of which was used for an independent sample labeling and hybridization reaction for each of the 28 classes of microdissected ovule samples (four stages for seven genotypes; Table 1, Appendix A) following the NimbleGen microarray labeling protocol of their One-Color DNA Labeling Kit. The labeled samples where then individually hybridized in random order, using the NimbleGen Hybridization System 12 to the custom *Ranunculus* arrays according to the producer’s instructions, and scanned using a NimbleGen MS 200 Microarray Scanner at 535 nm. Feature intensities were extrapolated using the DEVA software (version 1.1, Roche NimbleGen), and the raw expression data were normalized together using the DEVA implemented robust multiarray average (RMA) algorithm.

The normalized data were analyzed using the Qlucore Omics Explorer software (version 2.3). Principal Component Analysis (PCA) and Qlucore filtering by variance was implemented, whereby contigs with variation (σ/σ^max^) < 0.7 between apomictic and sexual groups were removed from the dataset. This threshold was chosen, using the Qlucore software, such that the differentially expressed apomictic and sexual probe groups could be visually separated on a PCA, while at the same time retaining the maximum number of differentially expressed genes. Second, the resultant set of differentially expressed genes was ranked according to *p*-value (<0.01) using a paired *t*-test between the apomictic and sexual groups at each stage separately. In addition to the *p*-value filter, log2 fold change >2 and adjustment of the *p*-value for multiple tests using a false discovery rate (FDR) with q-value < 0.05 were applied.

#### 2.2.4. Analyses of Gene Expression throughout Development

In order to calculate significant changes in gene expression patterns throughout ovule development between apomictic and sexual samples, the STEM software [51] was used to perform a similar analysis to that made by Sharbel et al. [31]. First, a data set of genes showing significant expression differences in at least one stage (but including the expression over all other stages) was constructed and used for STEM analysis. Except for the number of permutations, which was set to 1000 to increase accuracy [31], the profiling analysis was performed using default options with Bonferroni correction for multiple testing [51]. For the comparative analysis of different patterns of gene expression across the four developmental stages, the minimum number of intersected genes between sexual and apomictic samples was set to 1 [31] with maximum uncorrected intersection *p*-values < 0.05.

### 2.3. Analyses for Signatures of Ploidy, Parent of Origin Effects or Hybridization

Considering variable ploidy and evolutionary origins between sexual and hybrid apomictic *Ranunculus*, we sought to classify gene expression patterns into groups reflective of ploidy, parent of origin (sensu expression level dominance) or hybridization (sensu transgressive) effects. We first selected those genes showing statistically significant differential expression (*p* < 0.05, log2 fold change > 2, and FDR q-value < 0.05) between hexaploid apomicts and diploid sexuals, the two groups for which samples were available (Table 1) and hence for which statistical analysis was possible (due to sample size and lack of biological replicate, a similar statistical comparison using the tetraploid sample was not possible). Therefore, we used this subset of genes defined by the hexaploid apomict-diploid sexual comparison in all approaches. Then, in order to obtain information of partitioning of gene expression in the allopolyploid relative to each parent, we compared gene expression of each group, including data from the single tetraploid, estimating for each gene the log-fold expression difference of two contrasts: both parents to each other, and each parent to the allopolyploid, and controlling the distribution of *p*-values for each estimate using a false discovery rate of 0.05 [52].

#### 2.3.1. Ploidy

On a gene-by-gene basis, the standard deviation of expression was calculated for the hexaploid apomicts and diploid sexuals. Then, genes in the tetraploid and hexaploids whose expression was (1) lower than two standard deviations in the diploid, and (2) higher than two standard deviations in the diploid, were classified as exhibiting a statistically significant pattern reflective of ploidy/additivity effects (sensu Yoo et al. [53]). Thus, genes in the hexaploid were classified as displaying additivity and expression level dominance compared to either parent (for transgressive expression see Section 2.3.3).

#### 2.3.2. Parent of Origin Effects

Genes in sexual diploids and the tetraploid which (1) showed no expression differences with the hexaploid apomicts (i.e., <two standard deviations of mean gene expression), and (2) had expression levels which differed by >two standard deviations of the mean diploid sexual levels (the single tetraploid had no mean values per se), were classified as having patterns reflective of a parent of origin/expression dominance effect.

Lastly, we selected the group of all genes which did not show diploid-hexaploid parent of origin effects and were not initially selected (i.e., no significant differences in expression between hexaploid apomicts and diploid sexuals, see above). Next, we compared gene-by-gene the value of expression in the tetraploid against mean value of the hexaploid. Those genes in which the tetraploid sample had higher (i.e., >two standard deviations of the mean hexaploid apomict levels) or lower (i.e., <two standard deviations of the mean hexaploid apomict level) expression levels were classified as having patterns reflecting a tetraploid–hexaploid parent of origin/expression dominance effect.

In order to identify parental origin of alleles/factor(s) likely associated with apomixis (i.e., A^−^ putative genes), we compared high quality single nucleotide polymorphisms (SNPs) mined from a previous study [41] between the same individual plants analyzed here.

#### 2.3.3. Hybridization

In the opposite case, genes were classified as having a hybrid/transgressive pattern when they (1) showed no expression differences between diploids and tetraploid sexuals (i.e., <two standard deviations of mean gene expression), and (2) had expression levels which differed by >two standard deviations compared to the mean of hexaploid apomicts.

### 2.4. Microarray Validation Using qRT-PCR

Ten genes showing differential expression between apomictic and sexual genotypes across the four developmental stages were randomly selected (see Section 3). After retrieving the sequences from the assembled cDNA database used in the array manufacture, PCR primers were designed using the PrimerSelect software (DNASTAR Inc., Madison, WI, USA) and selected, when possible, to overlap the microarray probes with the following parameters: product size <150 bp, GC content between 40 and 60%, annealing temperature ca. 60 °C.

*Ranunculus*-specific reference genes were selected by identifying sequences which were homologous to a selection of *Arabidopsis thaliana* reference genes (www.tair.com) using a blast analysis (blastX 2.2.30+ using the default NCBI parameters). Based on maximum similarity (similarity > 95%, *e*-value < 1 e^−100^) homologous genes to UBQ (gb|ABH08754.1|) and ACTIN 11 (ref|NP_187818.1|) where chosen. Primers were designed following the procedures and parameters described above, and tested for amplification and expected product length in 10 μL PCR reactions including 25 ng of DNA, 1 μL of PCR Buffer II, 10 *p*mol for each primer, 0.025 U DNA Taq DNA Polymerase (Sigma-Aldrich, with 3.5 mM of MgCl_2_ and 4.95 μL of H_2_O. PCR reactions were performed in a Mastercycler ep384 (Eppendorf, Hamburg, Germany) using the following touchdown thermal cycling profile: 94 °C for 10 min; 9 cycles of 94 °C for 15 sec, 65 °C for 15 sec (1 degree decrease in temperature every cycle with a final temperature of 54 °C), 72 °C for 30 sec; 35 cycles of 94 °C for 30 sec, 57 °C for 15 sec, 68 °C for 2 min 30 sec; and a final 68 °C for 15 min.

qRT-PCR reactions, using UBQ and ACT11 (Dryad Id number: 82481 and 151955 Dryad entry doi:10.5061/dryad.nk151) as housekeeping genes, were run on a 7900HT FAST RT-PCR machine (Applied Biosystems, Foster City, CA, USA) using the SYBR Green Master Mix (Applied Biosystems) and the following program: initial denaturation at 90 °C for 10 min, followed by 40 cycles of 95 °C for 15 sec, and 60 °C for 1 min. Ct values (PCR cycle number where SYBR Green is detected) were extrapolated and used to infer initial copy number of the genes. Mean expression and standard deviation were calculated between two technical replicates and three biological replicates from apomictic and sexual genotypes using cDNA from the second stage of ovule micro-dissected tissues. Relative quantification was calculated using the ΔΔCt method and using the genes with higher CT values compared to the calibrator sample.

### 2.5. blastx, tblastx, Gene Ontology

Significantly differentially expressed genes obtained in all apomictic–sexual comparisons were selected for gene ontology (GO) analysis with Blast2Go (http://www.blast2go.com/b2ghome) using blastx (E-value cut off of E ≤ 1^−5^) and the default annotation parameters of the program. For genes where no blastx hits were obtained, an additional tblastx analysis was performed (E-value cut off of E ≤ 1^−5^). Overrepresentation analysis was not possible since only a small fraction of the *Ranunculus* transcriptome could be annotated (see Section 3).

## 3. Results

### 3.1. Gene Expression Differences

#### 3.1.1. Principal Component Analysis (PCA)

In order to recognize graphically possible subsets of genes that may account for the maximum distinction between the apomictic and sexual samples, PCA was applied to the normalized expression values of probes from the original 62102 contigs (≥300bp; raw data first normalized using the RMA algorithm implemented in the DEVA software). The resulting PCA graph showed a clear separation between samples with respect to ploidy, distinguishing apomictic and sexual groups (Figure 1).

#### 3.1.2. Stage Specific Differential Expression Analysis

Transcriptome-wide gene expression variation through ovule development was compared between sexual diploids and apomictic hexaploids of *Ranunculus*, whereby four pre- to post-meiotic ovule stages were compared between six different genotypes. Overall, across all stages 439 and 339 transcripts were found to be significantly down- or up-regulated between apomictic versus sexual genotypes (Figure 2; Venn A and Venn B, respectively). The distribution of up- and downregulated transcripts in apomicts differed across the studied developmental stages I to IV (30, 48, 247, and 3 upregulated, 44, 98, 58, and 27 downregulated for each stage respectively) (Figure 2).

Blast2Go analyses could be completed on only a small fraction of the differentially expressed transcripts, the limiting step being the inability to identify significant homologies to the NCBI nucleotide database(s). Of the number of genes which could be annotated (110 apomictically upregulated and 161 apomictically downregulated), only 59 (53.6%) upregulated and 78 (48.5%) downregulated transcripts from apomicts could be assigned a gene ontology (GO) term (Appendix A; Table 2). Hence, representation analysis was not possible due to the significant bias introduced by insufficient GO categorization. A full list of genes can be accessed at GenBank (BankIt2351891: MT624108 - MT624295).

A clustering-based STEM analysis [51] was performed to detect significant changes in patterns of transcript abundance across ovule development, by first grouping transcripts with similar expression profiles through development for the sexual and apomictic array datasets separately (Appendix A). In doing so, 46 and 62 transcripts could be assigned to three distinct patterns (STEM analysis, *p* < 0.01) in the sexual and apomictic groups, respectively. A comparison of patterns between these transcript sets identified eight transcripts as having significant differences (STEM analysis, *p* < 0.01) in the corresponding reproductive form, and showed a general trend of expression increase in developmental stage II followed by a sharp drop in stage III in apomicts (Appendix A). None of the eight genes could be assigned a GO term, and three had a significant homology to the transposon mutator sub-class protein (XP_006654086.1), salt overly sensitive 1b isoform 1 (EXC05020.1), and annexin-like protein (XP_007042996.1), respectively.

### 3.2. Transcriptome Wide Signatures of Hybridization, Ploidy Variation and Parent of Origin Effects

In order to understand and classify gene expression patterns of the apomictic hybrid (*R. carpaticola* × *R. cassubicifolius*) compared to the ancestral sexual parents (*R. carpaticola* and *R. cassubicifolius*), 304 differentially expressed transcripts were first identified in at least one developmental stage based on minimum fold change and statistical significance (log_2_ > 2, *p*-value < 0.01, FDR < 0.05) between the apomictic hybrids and the diploid sexuals (both groups having three genotypes each for statistical comparison). Secondly, a stage-by-stage comparison was performed for the expression of these 304 transcripts in the second tetraploid sexual parent (*R. cassubicifolius*; for which sample size precluded the first level statistical analysis—see Section 2) such that they could be classified into four different expression states (i.e., the hybrid relative to each ancestral parent). Genes showing expression bias with respect to the diploid or tetraploid parent were classified as showing a parent of origin effect, while those significantly over- or under-expressed in the apomict compared to both parents were classified as showing transgressive effects. Lastly, genes showing expression level changes in proportion to that of ploidy (e.g., increasing expression with increase of ploidy) were classified as showing a ploidy effect.

Examination of sequence similarity between genes showing a parent of origin effect and a high-quality SNP library (44) revealed only eight genes characterized by sufficient DNA sequence read coverage in all individuals, and of those only one (dyad contig id: 368515, Dryad entry doi:10.5061/dryad.nk151) showed 100% SNP similarity (for 16 SNPs) between the apomictic and the tetraploid genotypes only, implying a potential origin from the putative tetraploid parent (Appendix A).

The same procedure was applied to 19,116 genes that were not significantly differentially expressed between the apomictic hexaploids and the sexual diploids in order to classify an additional state of gene expression (2x-parent of origin; see methods). This approach was necessary considering that only a single 4x sample was used, and allowed the comparison of gene expression of this one parent (4x) to the hybrid apomict. This classification was further subdivided with regards to whether expression of the hybrid and the diploid were higher or lower in comparison to the tetraploid. In total eight different groups of genes were classified, with most being downregulated in apomictic ovules at stage II (Figure 3; Table 3, Appendix A).

Overall, most transcripts were expressed in a parent of origin pattern, followed by ploidy and then transgressive patterns (Table 3; Appendix A). The strongest effect was apparently due to parent of origin, with a total of 82 upregulated and 138 downregulated apomictic-specific transcripts. Ploidy-mediated patterns of expression were revealed in 62 and 93 apomictic specific upregulated and downregulated transcripts, respectively. Transcripts showing transgressive patterns were the least abundant, with 38 and 73 upregulated and downregulated in apomicts, respectively (Table 3).

Of all transcripts showing such patterns, only 83 (49 and 34 up- and downregulated in apomicts respectively) could be annotated (Appendix A). A simple screening of these genes revealed only three genes related to ovule development and reproduction. These three genes were coding for embryo sac developmental arrest protein (EDA), gamete expressed protein (GEX3), and a gene of the argonaute family (AGO) (e-value = 1.43 e-13, 3.28 e-32, 1.61 e-63, respectively and mean similarity of 47%, 70.5% and 84.4%, respectively; Appendix A). Other annotated genes were related to expression of retrotransoposons, transposon proteins, transcription factors, ribonucleases, kinases, among others (Appendix A).

### 3.3. qPCR Validation

Validation of the randomly selected genes showed concordance with the microarray analysis from the five upregulated and five downregulated genes. According to the REST software all 10 genes showed statistically significant up- and downregulation when measured with qRT-PCR (Appendix A).

## 4. Discussion

Whether caused by a single “master gene”, or through a polygenic complex, apomixis has long been a dilemma for scientists. Using different experimental approaches, the search for factor(s) underlying the switch in reproduction has led to the identification of a number of gene candidates in different species [22,54,55]. Such studies have also revealed a shift in gene expression through time for dozens or hundreds of genes normally involved in sexual reproduction [31,56,57]. Omic approaches have become useful for detecting expression shifts, although functional characterization and screening of the many candidates from such an experiment is a formidable task. This is especially true when dealing with wild species (as most apomicts are) in which annotated genomic information is not available. Additionally, the effects of polyploidy and hybridization [58] are often associated with the apomixis phenotype, adding an extra level of complexity to all analyses.

Despite hybridization and polyploidy introducing difficulties into data interpretations (see Mau et al. [59], their association with apomixis potentially reflects its mechanism of induction [18,60]. The “genomic shock”, i.e., the genomic perturbation at both genetic and epigenetic levels produced by the union of two different genomes [61,62], could trigger the cascade of spatial and timing mis-expression of sex specific genes and lead to apomixis [20,63]. More recent studies suggest that polyploidy is not essential for expression of aposporous apomixis, because it appears in diploid hybrids [37] and non-hybrids [64,65], albeit in low frequencies. These and other studies suggest that origin of apomixis in wild populations probably starts in diploid populations and is just indirectly enhanced by side-effects of polyploidy [66,67].

Continuing our ongoing work [41,44] with wild allopolyploid apomictic *Ranunculus auricomus* and related sexuals, we wanted to shed light upon the molecular basis for apomixis in *Ranunculus* and possible factors underlying the transition from sexuality to apomixis. We have analyzed genome-wide gene expression through morphologically-defined ovule developmental stages in both apomictic and sexual genotypes, and using a specific sampling strategy to maximize biological diversity which enabled the identification of differences encompassing both reproductive mode (e.g., apomictic versus sexual) and genetic background (e.g., different sample clusters) (see Figure 1). Thus, our approach allowed us to disclose species’ specific transcriptional variation connected to polyploidy and hybridity, and partition differentially expressed genes into patterns reflective of (a) reproduction specific expression, (b) heterochronic expression across ovule development, and (c) the expression of homologous genes in the apomicts and their ancestral (i.e., phylogenetic) parents.

### 4.1. Transcriptional Variation Reflect Contrasted Sexual vs. Apomictic Developments in the Ovule

In the *Ranunculus* cytotypes used here, Hojsgaard et al. [44] showed that the first stage of ovule primordium development is undifferentiated between apomictic and sexual *Ranunculus*. In contrast, diverse developmental irregularities accumulate during megasporogenesis, leading to perturbed and non-functional meiosis in apomicts (i.e., arrested development and/or altered megaspore selection; [44]), followed or accompanied by enlargement of a somatic cell that takes on the role of an aposporous initial (AI) cell. Deviations between the sexual and apomictic pathways continue during gametogenesis with subsequent development of the AI, parallel in timing with normal sexual gametogenesis in a few cases where meiosis progressed. The resultant unreduced megagametophyte (arising from the AI) then attains a similar multicellular structure to that of the sexuals at the final developmental stage [44]. Our tissue sampling design covered such developmental stages preceding and leading up to AI development till formation of female gametophytes (Table 1).

A reflection of the phenotypically-diverging development between pathways, gene expression differences between sexual and apomictic ovules show fewer differences in the total number of differentially expressed transcripts in the first and last stages of ovule sampling, whereas higher numbers of differentially expressed transcripts were found during the second and third stages (Table 2 and Figure 1). These findings agree with overall findings in other apomictic systems displaying global gene de-regulations in ovules at different developmental stages [68,69]. In our *Ranunculus* samples, stages II encompass AI appearance, usually in parallel with meiotic depletion, whereby the MMC did not enter meiosis or aborted during meiotic division. Stage III encompass the acquisition of a gametogenesis program whereby the AI enlarges to squeeze the forming meiotic megagametophyte (if any) and eventually take its place. Despite the relative weakness of any GO inference (see results), we note that no particular GO classes distinguished differentially expressed genes (Appendix A) at any stage. Especially in stage II, the appearance of aposporous initials and parallel development/arrestment of meiotic products (with almost equal frequencies between hexaploids [44]) may blur the distinction of genes expressed between these two cell lineages. Later on, in stage III, the differences between sexual and apomictic developments become most distinct (Figure 3) as further developmental competition increase rates of abortion in the meiotic pathway (whether of meiotic products, functional megaspore or early embryo sacs), and only unreduced gametophytes develop further. Stage IV might be principally not so different in gene expression as there are no key developmental transitions as in stages II (AI appearance, sporogenesis) and III (gametogenesis), and mature sexual and apomictic embryo sacs do not differ phenotypically.

### 4.2. Apomixis Inheritance in Allopolyploid Ranunculus and Partitioned Gene-Expression Effects

Apomixis involves changes to a number of individual (meiotic recombination, purging of mutations) and population (gene flow, adaptation) attributes which contribute to shape the life history of species [17]. As complex as the genetic processes underlying apomixis can be, it is believed to be controlled by one or few master regulatory genes [54]. In the 1980s, after a series of experimental crossings and offspring analyses, Nogler published a seminal work on the genetic inheritance of apomixis in *Ranunculus*. The association of apomixis and ovule gene de-regulation has been described in many plant species [22], thus, here we focus on discussing the relevance of our present results in the frame of Nogler’s ideas about apomixis inheritance, including more recent studies on different *Ranunculus* species.

#### 4.2.1. Homologous Parent of Origin Effects in the Apomictic Hybrid

As mentioned above, the hexaploid natural hybrid studied here originated during the last glaciation from a cross between a *Ranunculus carpaticola* diploid and a *R. cassubicifolius* tetraploid [41,48]. Our comparative analyses of gene expression patterns between each putative parental cytotype and its derivative hybrid lineage reveal taxon-related differences.

In the putative diploid parent, we found a relatively low number of transcripts with similar expression levels as in hexaploids, and thus the bias in gene expression of hexaploids toward the tetraploid parent suggests a parent of origin effect. Contributing to this effect is gene dosage, whereby the hexaploid is composed of two haploid genomes from the diploid parent *R. carpaticola* [48]. In regard to Nogler’s hypothesis, this observation suggests that A^−^ is absent (as expected) in diploids [35]. However, aposporous embryo sacs had been observed in very low frequencies in diploid *R. carpaticola* [44] which poses the alternative explanation that A^−^ is present in diploids but is not expressed, or that it is part of a multigenic network influencing the trait. Sequence similarity analysis using high quality single nucleotide polymorphisms (SNPs) mined from a previous study [41] revealed that only one of the genes showing a parent of origin effect and sufficient DNA sequence read coverage in all individuals showed 100% SNP similarity between alleles present in the apomictic and the tetraploid genotypes, but not to the diploid parent, implying a potential origin from the putative tetraploid parent. Considering the variability in read coverage between samples and the fact that the genotypes analyzed are, sensu stricto, not those involved in the original hybridization event circa 80,000 years before present [41], more cannot be inferred from the SNP dataset except that the diploid *R. carpaticola* probably lacks the factor A^−^.

Regarding the putative tetraploid parent, over all developmental stages 220 genes were found to be similarly expressed between 6x apomictic *R. carpaticola* × *cassubicifolius* and 4x *R. cassubicifolius* plants, once again suggesting a parent of origin effect and genomic dosage (6x has expectedly four putative haploid genomes from *R. cassubicifolius* [48]). Even though the relatively higher number of genes displaying a tetraploid parent of origin pattern of expression in *Ranunculus* ovules is likely biased by the lack of biological replicates, the result is concordant with similar studies in natural populations and synthetic hybrids of cotton [70], maize [71], rice [72] and *Senecio* [73]. All such studies showed similarly higher levels of parent of origin expression relative to transgressive effects, which tended to decrease with the age of the hybrid.

Investigations into the causes of parental expression bias point to the effect of *cis-trans* regulation, epigenetic changes, and introgression in organisms with different life histories like *Cirsium arvense* (thistle) [74] and cotton [53]. In *Ranunculus* synthetic hybrids (between diploids *Ranunculus carpaticola* × *notabilis* and tetraploid and diploid *R. cassubicifolius* × *notabilis*), apospory appeared spontaneously in diploid and triploid F_1_ hybrids of sexual species that did not previously show any signs of apomixis; however, functional apomictic seed were found in triploid F_1_ hybrids [44], and in diploid F_2_ hybrids [37], both in low frequencies.

Even though Nogler’s [35,36] apospory-incurring A^−^ factor is causal in the genomes of hybrids, it is plausible that such a factor is present in one (or both) parental species but not expressed. Previous morphological and genetic analyses of tetraploid *R. cassubicifolius* indicate that the cytotype is sexual [35,39,47,48], and embryological studies failed to show any sign of AI formation [44]. However, the high dosage of the wild-type A^+^ from the tetraploid sexual parent might explain the relatively high frequencies of sexual ovule (69%) and seed formation (29%) in hexaploid apomicts [44] compared to other polyploid apomicts having divergent evolutionary histories in which sexuality is found at residual levels (±5%, e.g., [75,76]). Likewise, *Ranunculus variabilis*, a naturally-related tetraploid apomict of the *R. auricomus* complex, showed overall higher frequencies of apospory and apomictic seed formation than our hexaploid hybrids used here [46]. These results suggest that the dosage of apomixis-factors depends on the parentage and evolutionary origin of plants, and does not necessarily increase linearly with the level of ploidy. Considering this, the origin of apomixis in the hexaploids could be explained in two ways. First, apospory could have appeared for the first time spontaneously in natural, triploid hybrids between a diploid sexual *R. carpaticola* and a tetraploid sexual *R. cassubicifolius* (as observed in 2x × 4x crosses mentioned above [44]). Subsequent polyploidization of the triploid hybrid could result in an hexaploid genotype like that established in nature.

Second, A^−^ could have already been present but not phenotypically expressed in tetraploid *R. cassubicifolius* plants. This possibility was already discussed by Nogler [35] as a probable consequence of dosage effects (see discussion below) and supported by observations on progenies from several rounds of backcrossing to the sexual parent which show a progressive delay in the initiation of apomictic development [35]. In many natural hybrids and polyploids, the correct spatial and temporal expression of developmental programs is altered, and is hypothesized as a central point for the development of apomixis [18,30,60,63]. Thus, the expression of apomixis in the hexaploids *Ranunculus carpaticola* × *cassubicifolius* analyzed here is likely associated with the observed heterochronic expression of genes during sporogenesis and gametogenesis which could only arise after hybridization or polyploidization.

#### 4.2.2. Ploidy Effect and Apomixis Expressivity in the Hybrid

Ploidy or additivity effects are due to the presence of multiple genome sets in pairs of diploid–polyploid species. Levels of genome and gene dosage can have relevant effects on different general plant attributes like heterosis [77], or reproductively related features like fitness [78]. However, disaggregating contributions of polyploidy versus hybridity to such attributes is usually not possible or require very specific experimental setups (e.g., using isogenic and hybrid genetic contexts [79]). In our analysis, the contribution of differentially expressed genes displaying apparent ploidy effects cannot be untangled from hybridity. Even though the relative contribution of ploidy effects to the total number of differentially expressed genes is intermediate (below those grouped under a parent of origin effect; Table 3), together with transgressive effects (hybridization) they contribute the most genes to the molecular changes driven the emergence of apomixis in *Ranunculus* ovules. At this point, the number of differentially expressed genes is not high enough to suggest global heterochrony rather than a few dominant factors is the molecular basis for apomixis in this group. Therefore, we will focus our discussion on possible genome dosage effects associated with ploidy and hybridity, and framed by Nogler’s theoretical model of apomixis inheritance.

In his cytological and inheritance studies in *Ranunculus*, Nogler [36] further observed that in experimental crossings increased dosage of the controlling factor for apomixis (A^−^) relative to the wild type sexual A^+^ allele resulted in higher levels of apomixis penetrance, as evidenced by the occurrence of AI cells. Results of Barke et al. [37] on diploid hybrids also support an allelic dosage effect. Assuming that A^−^ is allelic (or epiallelic), it then becomes relevant to understand how dosage could vary (see [35]) between cytotypes. Tetraploids are fully sexual [44], implying either that they lack A^−^ altogether (and hence A^−^ must have appeared in hybrids only), or that A^−^ is present in tetraploids but because of wrong timing (see above), low dosage (A^−^ A^+^ A^+^ A^+^ genotypes) and/or lack of a hybrid background, it is not expressed (i.e., low or lack of AI cells, as observed in natural diploid and tetraploid sexual *Ranunculus* spp. and experimental hybrids [37,44]).

In hexaploid apomicts, one or more copies of A^−^ in a hybrid background could result in increased expression levels with respect to the diploid and tetraploid parents of lower ploidy. In fact, both the frequencies of AI’s (67%, ranging between 50–87%) in hexaploids and observed variability in expressivity of apomixis (71% ± 12%) [44] suggest variable dosage of factors directly or indirectly related to apomixis (i.e., A^−^). The observed variability in levels of sexuality in these facultative hexaploids (mean value 69%, ranging between 56–96% of ovules with functional megaspores [44]) point to hexaploids being heterozygous for A^−^ and the wild type factor A^+^. Since *Ranunculus* plants were grown under identical conditions, such variability may be due to a variable dosage for A^−^ and/or distinct genomic contexts. Multiple copies of A^−^ can be acquired by gamete recombination and formation of sexual offspring between facultative hexaploids. However, an increase in copies of A^−^ would be restricted by its dosage effects, as increasing expressivity (i.e., level of apomixis) will concomitantly reduce levels of residual sexuality, thus making it increasingly impossible to attain an obligate (100% expressivity) apomictic hexaploid individual via recombination. This explains observations in *Ranunculus*, *Paspalum* and many other apomictic genera in which no fully apomictic plants had been so far recovered ([46,65,80,81]; although, see [82] for cases of obligate apomixis in *Boechera*). The opposite situation is also possible, with occasional recombinant individuals having no copies of A^−^ that have lost their capacity to reproduce apomictically. Even though it has not been observed in natural apomictic hexaploid *Ranunculus auricomus* individuals [44,45,46,48], a reversal to sex in an apomictic lineage has been rarely observed [81,83] but is theoretically expected in apomictic populations [67,84,85].

From a phylogenetic viewpoint, in the species-rich *Ranunculus* genus most taxa have unique biogeographic and phylogenetic histories [86]. Apomixis appeared at least twice independently [39]. Hence the origin of the A^−^ allele in independent lineages can certainly be of diverse nature (i.e., via vertical or horizontal transfer, through polyploidy and/or hybridization). Considering our pedigree system of diploid and tetraploid parents and derivative hexaploid hybrid, we have partial evidence that the A^−^ factor originating apomixis in hexaploids was inherited from the putative tetraploid parent. Yet, one cannot assume with certainty whether A^−^ is absent in the diploid or polyploid plants studied here, whereby they display apomixis elements but lack functional apomixis (e.g., low levels of multiple embryo sacs where observed in diploid sexual *R. carpaticola*; [44]). In any case, expression patterns of A^−^ and its dosage would reflect ploidy effects, being absent/not expressed in diploids, lowly expressed in triploid carriers and highly expressed in the hexaploid apomict (as observed in Hojsgaard et al. [44]).

#### 4.2.3. Transgressive Gene Expression in the Apomictic Hybrid

Transgressive segregation relates to the formation of extreme or transgressive phenotypes (falling beyond the parental range) often observed in segregating populations mainly for traits influenced by multiple quantitative loci (QTL) [87]. Studies on apomixis inheritance have demonstrated that apomixis factors deviate from Mendelian segregation due to modifiers and epistasis associated with secondary factors [32]. In our analysis, observed transgressive gene expression effects may be influencing the dominance of A^−^ (see discussion below), and the extent to which such transgressive expression attenuates apomixis expression would require another experimental design.

Expression patterns of an apomixis factor A^−^ and associated cofactors could, in addition to the parent of origin and ploidy effects discussed above, be associated with *cis* and *trans* regulatory dynamics that characterize hybrid genomes [88]. Aposporous initials have been observed in the tetraploid *R. megacarpus* carrying the postulated A^−^ factor [35], and in a low percentage in synthetic diploid and triploid hybrids between sexual parents [37,44]. Although A^−^ might control aposporous embryo sac formation as have been described in other aposporous systems [89], A^−^ cannot directly control the timing of AI induction as it is dosage dependent [35]. Thus, since the timing of AI induction is key for successful apomictic development [35], transgressive gene expression caused by genome merging during hybridization seems to have a relevant role in *Ranunculus*. Transgressive gene expression effects might well cause shifts in timing of floral development and reproductive programs underlying developmental heterochrony and the transition to apomixis in hybrids [18]. The identification of 35 and 38 homologs showing transgressive effects which were down- and up-regulated in the apomictic, respectively, support this view. Such a relatively low number of genes displaying transgressive expression could be a reflection of the old evolutionary age of the hexaploid lineage (about 80,000 yo [41]), as suggested by Hegarty et al. [73] whereby transgressive effects caused by genomic shock after hybridization are ameliorated by genetic mechanisms and evolutionary forces acting upon neopolyploids during first generations post-hybridization and polyploidization.

Despite differentially expressed genes falling in the transgressive expression effects group contributed the less to the overall gene dysregulation in hexaploid *Ranunculus*, the only three genes related to plant reproduction out of all genes that could be annotated were from this group. This also suggest transgressive rather than ploidy or parent of origin expression effects might play a more active role on apomixis emergence. These genes are involved in megagametogenesis and development, ones being the embryo sac developmental arrest protein (EDA), gamete expressed protein (GEX3), and a gene of the argonaute family (AGO). Interestingly, *Arabidopsis* EDA mutants show a series of defects during megagametogenesis, resulting in interrupted or abnormal meiotic division [90]. Furthermore, GEX3 is essential for pollen tube guidance during double fertilization, and its misregulation in *Arabidopsis* resulted in reduced seed set and undeveloped embryos [91]. If these genes had analogous phenotypic effects in *Ranunculus*, its transgressive expression in hexaploid apomicts is likely modulating sexual vs. apomixis expression and seed formation, which might explain observed developmental variability [44,46]. Finally, in *Arabidopsis* and Maize genes defective for AGO9 and AGO104 were shown to control female gamete formation via a small RNA pathway controlling methylation and transcription of many targets in the ovary [92,93]. Mutants displayed apomixis-like phenotypes including AI-like cells and unreduced gametes [92,93]. In *Ranunculus* in general and in the hexaploid apomictic *R. carpaticola* × *cassubicifolius* plants, the AGO gene might also be associated with a possible epigenetic control of apomixis and gamete formation. Likewise, observed variability of proportions of facultative apospory under different light stress conditions in hexaploid clone-mates of *R. carpaticola* × *cassubicifolius*, including residual levels in diploid and tetraploid cytotypes, suggests an influence of epigenetic regulatory mechanisms on reproductive phenotypes [45,46]. In *R. kuepferi*, differential cytosine-methylation patterns found in sexual and apomictic natural populations support this hypothesis [94]. In other apomictic plant groups different genes had been found linked to the apomixis phenotype or to individual components of apomixis (e.g., [95,96,97,98,99,100,101]). Even when these three genes might have important roles in the functional expression of apomixis in *Ranunculus*, it seems more likely they are being involved in the molecular cascade rather than being master genes associated to apomeiosis, parthenogenesis or endosperm development.

Global shifts in gene expression patterns associated with hybridization and polyploidy [102] have been hypothesized to underlie the switch from sex to apomixis [18]. Experimental support for this hypothesis has been found in apomictic *Boechera* [31,103] and between sexual species of *Tripsacum* [30]. The hexaploid apomictic *Ranunculus* lineage analyzed here had a hybrid origin around the last glacial maximum [41,47,48]. Hybridization as a trigger of apomixis in this lineage is supported by synthetically-derived hybrid *Ranunculus* mirroring the original hybridization event which show formation of apomeiotic embryo sacs and low rates of functional apomixis in the first two generations after hybridization [37,44]. This hypothesis is supported by our gene expression analyses in sexual and apomictic ovules pointing to at least three genes likely associated with the expression of the apomixis phenotype, with influences of hybridity and polyploidy reflected in dosage and transgressive effects. Similar reproductive phenotypes have been observed in sexual mutants [92] and other experimental hybrids in *Sorghum* and *Antennaria* [21]. Interestingly, despite having different forms of apomixis [104], diplosporous *Boechera* [31] and aposporous *Ranunculus* similarly show a negative spike in differential gene expression at similar stages of megaspore mother cell/aposporous initial cells progression during ovule development in comparisons between sexual–apomictic ovaries. Even though such parallelism on gene expression changes at key developmental stages may point to conserved processes leading to apomeiosis, specific studies will be required. Genetic and functional evaluation of EDA, GEX3, and AGO orthologues in *Ranunculus* will shed light on regulatory mechanisms through which apomixis and sexuality are modulated in polyploid plants.

## 5. Conclusions and Prospects

In this study we investigate gene expression variation in the apomictic *Ranunculus auricomus* complex and present a list of genes that show differential expression between aposporous and sexual ovules across four developmental stages. Considering the common difficulty in wild species for which genomic sequence information can be readily collected but little to no gene annotation can be made, we have considered the natural history of the *Ranunculus auricomus* complex in order to classify genes whose expression patterns reflect the evolutionary and genetic mechanisms hypothesized to underlie the switch from sex to apomixis. Despite this, our study cannot confirm a molecular basis for apomixis based either on global heterochronic gene expression or dominant genetic factors. The results suggest apomixis in *Ranunculus* might depend on changes of reproductive genes with downstream influence on different gene regulatory cascades. Thus, our observations based on the number of differentially expressed genes and their expression patterns are in concordance with the commonly accepted idea that apomixis is caused by gene de-regulation of the sexual reproductive program in connection with hybridization and polyploidization. Here we set the foundations for more specific studies on gene regulation with respect to apomixis expression and expressivity.

## Figures and Tables

**Figure 1 genes-11-00728-f001:**
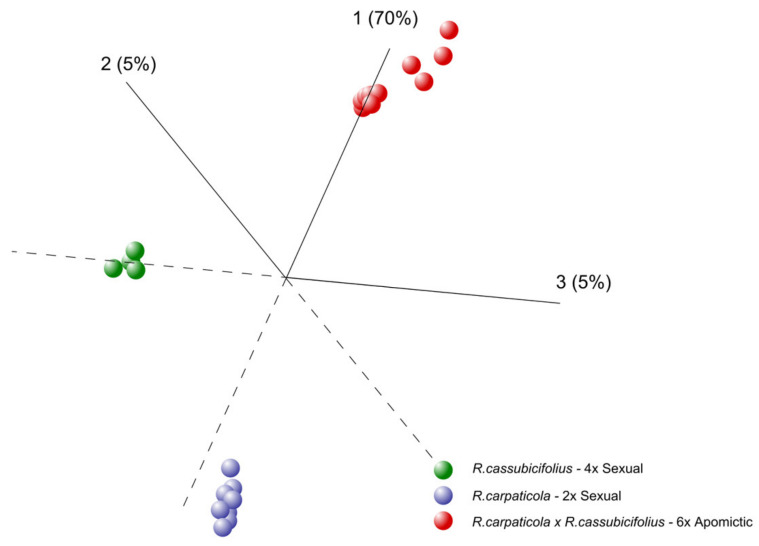
Principal component analysis (PCA) of gene expression on apomictic and sexual *Ranunculus*. PCA applied to normalized microarray data representing 4 ovule developmental stages from 3 apomictic (red) and 4 sexual (blue and green) *Ranunculus* genotypes showing ploidy and reproduction specific effects. Each dot represents one ovule stage, and frequencies in parentheses show the percentage of total variation explained by that principal component. The numbers on the axes represent the respective component (and the % of variation for each component).

**Figure 2 genes-11-00728-f002:**
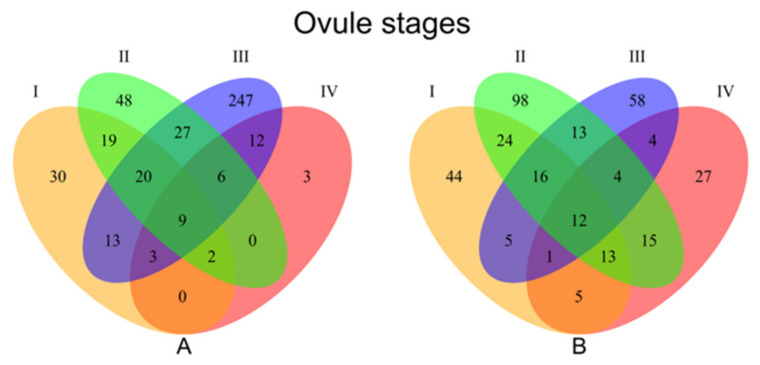
Venn diagrams showing number of differentially expressed transcripts between apomictic and sexual genotypes at each ovule developmental stage. A: Genes downregulated in the 6x apomicts compared to 2x sexuals B: Genes upregulated in the 6x apomicts compared to 2x sexuals. I = premeiotic, II = meiotic/aposporous, III = early embryo sac, IV = mature embryo sac.

**Figure 3 genes-11-00728-f003:**
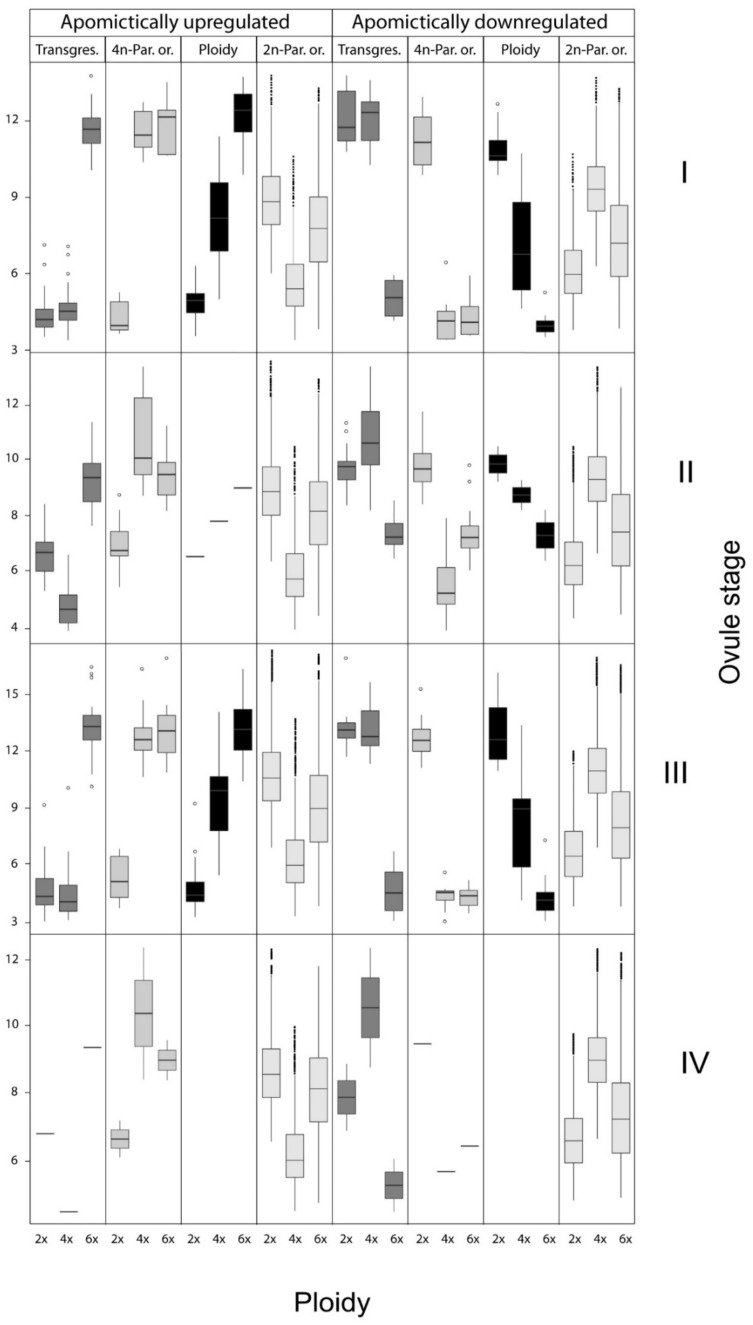
Box plots of expression distributions for genes which were significantly up- or downregulated in at least one apomictic ovule stage, as grouped into transgressive, ploidy-mediated and parent of origin patterns.

**Table 1 genes-11-00728-t001:** Diploid, tetraploid and hexaploid *Ranunculus* samples used in this study.

Taxon (Code)	*N*ind	Ploidy ^1^	Reprod Mode	Apospory Expressivity ^2^	Collection Locality, Collector, Collection Number, and Vouchers
*R. carpaticola* REV1	3	2x	Sex	0.0	Slovakia, Slovenské rudohorie, Revúca, hill Skalka (forest) Hörandl 8483 (WU)
*R. carpaticola* × *cassubicifolius* TRE	1	6x	Apo	0.226	Slovakia, Strázovské vrchy (near Trençín), between Kubra and Kubrica (margin of forest and meadow) Hörandl et al. C29 (SAV)
*R. carpaticola* × *cassubicifolius* VRU 2	2	6x	Apo	0.339	Slovakia, Turçianska kotlina, Vrútky-Piatrová (meadow) Hörandl et al. C35 (SAV)
*R. cassubicifolius* YBB 1	1	4x	Sex	0.0	Austria, Lower Austria, Wülfachgraben, SE Ybbsitz (forest) Hörandl 8472 (WU)

^1^ For ploidy identification see [47,48]; ^2^ Data from [44]. WU: herbarium of the University of Vienna, SAV: herbarium of the Institute of Botany, Slovak Academy of science, Bratislava.

**Table 2 genes-11-00728-t002:** Numbers of differentially expressed transcripts at each of four ovule developmental stages.

Number of Contigs	Number of Contigs, >300 bp	Differentially Expressed Transcripts			Annotated
462102	62102				
		Develop. stage	Upregulated ^1^	Downregulated ^1^	
		I	120 (44)	96 (30)	72
		II	195 (98)	131 (48)	107
		III	113 (58)	337 (247)	156
		IV	81 (3)	35 (3)	30

^1^ data from 6x apomictic individuals compared to the normal expression in 2x sexuals; numbers in brackets represent number of stage specific transcripts.

**Table 3 genes-11-00728-t003:** Number of differentially expressed genes ^1^ associated with each ovule developmental stage and expression class in diploid–polyploid comparisons (brackets show relative percentage compared to the overall stage number of genes).

Expression Class ^2^	Develop. Stage	Transgressive Effect	Parent of Origin Effect	Ploidy Effect	Total
Apomictic 6x upregulated ^3^	I	5 (0.1)	27 (0.49)	23 (0.41)	55
	II	15 (0.58)	10 (0.38)	1 (0.04)	26
	III	16 (0.16)	44 (0.46)	36 (0.38)	96
	IV	2 (0.4)	1 (0.2)	2 (0.4)	5
	Total up	38 (0.21)	82 (0.45)	62 (0.34)	182
Apomictic 6x downregulated ^3^	I	6 (0.26)	7 (0.3)	10 (0.44)	23
	II	18 (0.32)	37 (0.65)	2 (0.03)	57
	III	9 (0.24)	11 (0.3)	17 (0.46)	37
	IV	2 (0.4)	1 (0.2)	2 (0.4)	5
	Total down	35 (0.29)	56 (0.46)	31 (0.25)	122

^1^ for seq. names, GO terms, enzyme codes and statistics refer to Appendix A; ^2^ compared to normal sexual diploid expression; ^3^
*p* < 0.001, FDR < 0.05, log_2_ change > 2.

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
