# Peer review of "Chasing the Apomictic Factors in the *Ranunculus auricomus* Complex: Exploring Gene Expression Patterns in Microdissected Sexual and Apomictic Ovules"

_genes, 2020, doi:10.3390/genes11070728_

Round 1

Reviewer 1 Report

Pellino et al. “Chasing the apomictic factors in the Ranunculus auricomus complex: exploring gene expression patterns in microdissected sexual and apomictic ovules”

In this manuscript, the authors conduct transcriptomic analysis of microdissected ovules of both sexual and asexual taxa in the R. auricomus complex, a model system for the study of apomixis in flowering plants. They explore hypotheses for the genetic basis of apomixis using comparative transcriptomics, and assess expression patterns as related to parent of origin effects, ploidy effects, or hybridization (transgressive) effects. No apomictic model system is uncomplicated, and I think that this approach is a creative and well thought out approach to addressing the traits that are commonly associated with apomixis. I think this study is absolutely ready for publication, and I look forward to citing it in the future. With that said, I have a couple of major comments related to the framing and methods of the paper.

Major comments

The main issue I have with the manuscript is one of framing, and how the data available address the question posed by the authors. I believe that the question driving this study is: what is the genetic basis of apomixis in this species complex? This is of course a large question that can only be answered through the accumulation of many studies using a variety of approaches, so although I do not believe the authors have answered this question here, I certainly did not expect that question to be answered. But after reading the manuscript, I’m not sure that the collective framework of Nogler’s A+ and A- factors vs. the heterochrony hypothesis are really addressed by the ploidy/parent of origin/transgressive classification system. I understand that this framework is necessary to account for the complexity of apomixis expression in the system, and I believe it is the right way to approach the data, but it is difficult to draw conclusions about Nogler’s factors and heterochrony/dysregulation from the framework as it currently stands. I think the manuscript would benefit from some additional text in the introduction that addresses the difficulty of disentangling these hypotheses given the complexities of the system. For example, some mention of how these hypotheses can be differentiated using RNAseq data. I also found the discussion, although very complete, to be difficult to follow. I believe it would benefit from a restructuring of the existing text to highlight what we have learned about the dominant factors vs. dysregulation hypotheses from each expression category (parent of origin, ploidy, and hybridization) rather than the other way round.

My other issue is that I wish the authors had additional samples of the tetraploid parent, as I think that may contribute to some bias in the results. However, this is outside the scope of this manuscript, and I do not think that issue should delay its publication.

Minor comments

Lines 497-506: The SNP study mentioned on lines 497-506 does not appear in the Methods section, and I believe it should. It’s also not entirely clear to me how the authors conclude that the diploid parent likely lacks the A- factor. I think that would be easy to clarify with a sentence or two.

Line 182: I’m a bit concerned that ovules were collected over multiple days, as I wonder if the level of apospory in the asexual accessions changes over the flowering period, as reported for other taxa. I’d be interested to know what the authors' response is to this.

Lines 256-261: I believe this is the section that refers to the 2x parent of origin section in Table 3, but it would be helpful if the authors clarified this in the text, so that the reader may easily link the two.

Lines 352 & 354: I’m wondering what kind of statistical test these p-values come from. More information on the tests should be added to the parentheticals.

Line 467: I’m not entirely sure that I understand what is meant by “blurring the distinction”. A bit more clarification on this (i.e. is the frequency of AIs and meiotic products the issue?) would be helpful.

Figure 1: I can easily see that the groups are distinct, but it’s difficult to see from the figure where these groups lie along the axes. I think the figure would be easier to interpret if additional views of the rotation were included, or if the axes were part of a cube representing the three dimensions.

Figure 3: This figure contains a lot of information, presented in a clear way. However, I did find it difficult to read the Y axes, so I recommend increasing the font size and perhaps standardizing the axes if possible. The variation in line weights separating the subcomponents also contributes to a bit of visual confusion. I think in general that presenting a figure showing expression variation is great, but box plots may not be the best way to present this data—as exemplified by the plots that contain only one datapoint in some categories (i.e. Stage IV transgressive upregulated). Violin plots may be more informative for the reader.

Table 3: This table is clearly laid out and informative. I suggest adding totals at the bottom for each sub-category (perhaps with and without the apomictic-2x sexual comparisons for the parent of origin effect sub-category).

Figures S1 and S4 do not appear in the manuscript. Figure S4 is not listed in the supplementary material.

There are also some grammatical issues in the introduction and discussion that should be resolved prior to final publication.

Author Response

Major comments

Rev 1: The main issue I have with the manuscript is one of framing, and how the data available address the question posed by the authors. I believe that the question driving this study is: what is the genetic basis of apomixis in this species complex? This is of course a large question that can only be answered through the accumulation of many studies using a variety of approaches, so although I do not believe the authors have answered this question here, I certainly did not expect that question to be answered. But after reading the manuscript, I’m not sure that the collective framework of Nogler’s A+ and A- factors vs. the heterochrony hypothesis are really addressed by the ploidy/parent of origin/transgressive classification system. I understand that this framework is necessary to account for the complexity of apomixis expression in the system, and I believe it is the right way to approach the data, but it is difficult to draw conclusions about Nogler’s factors and heterochrony/dysregulation from the framework as it currently stands. I think the manuscript would benefit from some additional text in the introduction that addresses the difficulty of disentangling these hypotheses given the complexities of the system. For example, some mention of how these hypotheses can be differentiated using RNAseq data. I also found the discussion, although very complete, to be difficult to follow. I believe it would benefit from a restructuring of the existing text to highlight what we have learned about the dominant factors vs. dysregulation hypotheses from each expression category (parent of origin, ploidy, and hybridization) rather than the other way round.

We have added a paragraph in the introduction addressing the challenge of disentangling between simple genetic factors regulating apomixis and heterochronic development (lines 121-127).

In addition, we have partially restructured the discussion accordingly, in an attempt to put our ideas in a clearer perspective, and we have included a few comments (see lines 607-609, 710-716 and 744-747) highlighting the current knowledge about dominant factors vs. dysregulation. We do understand separating our debate about Nogler´s hypothesis into one subsection would benefit to some extent the discussion; however, in doing so we will need to repeat parts of the discussion which would also add some complexity to the text. For this reason, we would prefer keeping the paragraphs about Nogler´s hypothesis in each subsection while considering specific expression effects. However, if the reviewer insists, we will separate Nogler´s discussion into one subsection.

Rev1: My other issue is that I wish the authors had additional samples of the tetraploid parent, as I think that may contribute to some bias in the results. However, this is outside the scope of this manuscript, and I do not think that issue should delay its publication.

Yes, we do agree on this. We would like to have additional tetraploid samples. Unfortunately, this was not possible during the development of the experiments, for different reasons, the most relevant was the lack of enough plant material and plants at similar developmental stages. To avoid confusion in interpretation of results, we have moved the data of the tetraploid sample out of the Table 3 and included it as Suppl. Table.

Minor comments

Rev1: Lines 497-506: The SNP study mentioned on lines 497-506 does not appear in the Methods section, and I believe it should. It’s also not entirely clear to me how the authors conclude that the diploid parent likely lacks the A- factor. I think that would be easy to clarify with a sentence or two.

We have now placed details on this analysis in Methods (lines 280-283) and Results (lines 396-400). We do not conclude the diploid parent lack the factor A-, but rather point to a factor coming from the tetraploid parent that is differentially expressed and show 100 % similarity to a gene in the 6x, suggesting it might have a role in apomixis.

Rev1: Line 182: I’m a bit concerned that ovules were collected over multiple days, as I wonder if the level of apospory in the asexual accessions changes over the flowering period, as reported for other taxa. I’d be interested to know what the authors' response is to this.

According to previous embryological and cytological analyses in ovules and seeds of the same genotypes used here, either the number of initial of apospory cells (usually 1-2) or the amount of sexual vs. asexual seeds change in a short time period. We had made the ovule extractions in a two weeks time window. This is now clarified in the text (line 141). In addition, the levels of sexuality and apomixis were discussed in the text (lines 623-629).

Rev1: Lines 256-261: I believe this is the section that refers to the 2x parent of origin section in Table 3, but it would be helpful if the authors clarified this in the text, so that the reader may easily link the two.

The paragraph was rewritten (see lines 267-282).

Rev1: Lines 352 & 354: I’m wondering what kind of statistical test these p-values come from. More information on the tests should be added to the parentheticals.

We used a STEM analysis which is based on a STEM clustering algorithm. WE added this information in the text (see lines 370, 373-376).

Line 467: I’m not entirely sure that I understand what is meant by “blurring the distinction”. A bit more clarification on this (i.e. is the frequency of Ais and meiotic products the issue?) would be helpful.

We refer to genes expressed by one or the other cell lineages (i.e. meiotic or apomictic) when they occur together in the same ovule. We have now clarified this point (line 501).

Figure 1: I can easily see that the groups are distinct, but it’s difficult to see from the figure where these groups lie along the axes. I think the figure would be easier to interpret if additional views of the rotation were included, or if the axes were part of a cube representing the three dimensions.

A new Figure 1 is included.

Figure 3: This figure contains a lot of information, presented in a clear way. However, I did find it difficult to read the Y axes, so I recommend increasing the font size and perhaps standardizing the axes if possible. The variation in line weights separating the subcomponents also contributes to a bit of visual confusion. I think in general that presenting a figure showing expression variation is great, but box plots may not be the best way to present this data—as exemplified by the plots that contain only one datapoint in some categories (i.e. Stage IV transgressive upregulated). Violin plots may be more informative for the reader.

We have improved Figure 3.

Table 3: This table is clearly laid out and informative. I suggest adding totals at the bottom for each sub-category (perhaps with and without the apomictic-2x sexual comparisons for the parent of origin effect sub-category).

Done. See Table 3.

Figures S1 and S4 do not appear in the manuscript. Figure S4 is not listed in the supplementary material.

Yes. Thank you. We have mistaken Figure codes. This is now amended.

There are also some grammatical issues in the introduction and discussion that should be resolved prior to final publication.

The Introduction and Discussion sections were checked, rewritten in some parts, and typos were corrected.

Reviewer 2 Report

Pellino et al. performed a comparative transcriptomic study of apomictic and sexual Ranunculus. They used four different developmental stages and microdissected the ovules. They use microarrays as a technique which is surprising since the authors have already published transcriptomic read data for the system which is more state-of-the-art. Their analyses identify sets of differentially expressed genes between sexuals and apomicts and attempted to make inferences on the “expression effects” ie. effects of ploidy, parent-of-origin and hybridity on gene expression.

A main concern is the apparent lack of proper experimental design for a transcriptomic study that aimed to identify patterns of differential gene expression. It is widely accepted in the community the use of at least 3 biological replicates per genotype for proper statistical analyses to be performed. This is not the case in this study. Table 1 discloses the samples used and clearly there are 7 individuals that were sampled for 4 stages for a total of 28 samples. The only sample that would have the proper replicates in this case is REV1. Now the authors argue in l. 364 that the comparison of apomictic hybrids and the diploid sexual with both groups having three genotypes each for statistical comparison is valid. This may be accepted but of course the reality is that the hexaploidy apomicts are actually two genotypes and one is not replicated. This biological variance may affect the accuracy of results. Added to the lack of proper replication is the fact that differences in expression may be due to the fact higher ploidy level when comparing the hexaploid apomicts to diploid sexuals and the genomic and transcriptomic consequences of that biological factor rather than anything to do with apomixis as such. This also can be accepted as polyploidy and apomixis are related and with transcriptomic studies as this, it is impossible to disentangle its different effects on the gene expression patterns. Care should be taken with conclusion statements and address these uncertainties underlying the grounds for differences in expression.

Also regarding statistical soundness of the study, it was difficult for me to follow the section “Analyses for signatures of ploidy, parent of origin effects or hybridization”. I kindly request the authors their clarifications. The authors concede the lack of statistical power they have to draw any significant inferences on the role of the tetraploid parental R. cassubicifolius YBB 1. Indeed, any conclusion from this analyses is highly doubtful as there was a SINGLE individual sampled. No replicates at all. I would recommend to refrain to use this sample at all. Furthermore, the methodology implemented for the “expression effects” relies heavily on comparisons and filtering of datasets to expression patterns on this single tetraploid sample, for example as described inlines 247, 252, 258. This type of analyses is tipically done by using the hybrid expression estimates (in this case the hexaploid apomict) and comparing it to the parentals, here the diploid and tetraploid sexuals. In the current analyses filtering and sorting is done by using the tetraploid parent and this seems counterintuitive for me. Please let me know what am I getting wrong here. This needs clarity for further readers and the issue of the lack of replicates of the tetraploid parent needs to be addressed. Another concern is the results per se. Formatting of figures is an issue. An important contribution of studies as this is the identity of the transcripts identified in the analysis of differential gene expression and here, those of “expression effects”. These results summarized in table 3 and some supplementary tables are far from informative and most importantly lack the information that would allow researchers to pinpoint these genes in Ranunculus individuals for future studies. It is imperative that these transcripts are properly identified and made traceable for the scientific community. The authors have already published raw read data and fasta sequence for the transcripts. It would be easy for them to match the genes identified in their analyses sets to a fasta sequence to be deposited in genebank or to be traceable to the transcriptome assembly they have already published. Also, regarding the suggestion on improving annotation reporting and results in general. On the very last part of the discussion the identity of three “choosen” genes is disclosed and discussed. Part of this should be move to results and I suggest to expand on interesting gene candidates that resulted from all analyses.

I think that although not an ideal experimental design for the objectives set for this manuscript the dataset is valuable and does addressed an important question in an interesting biological system, with putative parental samples and a polyploid hybrid apomict. I believe the comparisons of sexual diploid vs. apomicts are sound (each with three replicates) and those results are accurate. The identities of the genes as well as their inheritance patterns in relation to ploidy and hybridity are an important contribution (although not properly reported at the moment) and worth of publication. I hope my suggestions will help in the improvement of this manuscript.

Hereby more detailed comments on the document contents:

  • L 170. The authors declare multiple technical replicates for each gene, but these are all probes that bind to the same original cDNA molecule provided. These are therefore different readings of the same molecule, I would therefore not consider this a technical replicate. There is a single molecule that was
  • L 181. Sup. Figure 1 does not show any details of the collection stages.
  • L 205 - figure s3 shows results, remove or clarify
  • L 219 you performed paired t test between sexual and apomicts grouping different genotypes and ploidies in those groups. Each of those samples is an independent genotype that should not be grouped with other samples. I think this analyses are statistically faulty. One wants to dissect the effects of apomixis but here ploidy as well as genotype will affect expression estimates and this is not accounted for. More appropriated analyses ie. Nested designs including genotype and ploidy as factors, could be implemented.
  • 2.4 is difficult to follow. In particular line 227. I understand a gene set is selected based on DGE results. These DEG are selected based on a per stage criteria, as explain in 2.2.3. and then “the expression over all other stages was added to the data set”. Please clarify.
  • On results it is then clear the STEM analyses are made on apomicts and sexuals separately. This should be described the methods section.
  • L 231 please shortly explain what are the statistics you use in your analyses, the citation of Rapp et al is vague
  • L 232, reports using a uncorrected p value for analyses. Please argument why you don’t use the FDR correction value?
  • 249 The use of the tetraploid data and its approximation is statistically faulty the authors claimed that by setting their standard deviations limits is “statistically significant” and I do not think it is, as no test has been performed for comparisons involving the tetraploid samples.  
  • 250 please shortly elaborate on the classifications described in the reference Yoo et al.
  • 264 the comparisons between the sexual samples is not statistically explained in methods. Sample size differs (3 to 1 individuals in table one, cannot account for replication). Nowhere is explained how this test for differential expression is performed. If it is just standard deviations limits, and with these differences in sampling I again claim that the conclusions drawn from these analyses are statistically faulty.
  • 5 I find it unfortunate that the annotation job was done so poorly. It is very uninformative to see the supplementary tables filled with NA fields. I find that E ≤ 1-5 is a high threshold and I would regardless of e-value report the top hit for all transcript identified, (and all relevant blast results) and highlight those with a high confidence (ie. E ≤ 1-5) match. Even if no hit would be reported, the fasta sequences for this genes identified in the differential gene expression analyses as well as the genes identified with ploidy, parent of origin and hybridity effects. They should be easily recovered from the reference transcriptome produced and these transcript sequences are a very important part of the results that should be made available for the community in general.
  • L 309 – lack of clarity. How do you identify a subset of genes with a PCA analyses on overall sample variation? Please elaborate on this objective.
  • L338-344. What is the objective of making these GO representation/assignment analyses? If the representation analyses does not have power and does not make a contribution to the biological question addressed in this study I suggest to remove this.
  • Figure 3 is almost illegible. Needs complete reformatting, probably to be splitted?. Significance test needs to be performed between the boxplots distributions. sample size (n) should be disclose above each box. Etc.
  • 497-502- SNP validation of parent of origin effects are not reported in Methods. This should actually be in methods and results and not squeezed in Discussion.
  • There is an extensive part (525- 550, all of 4.2.2 part on ploidy differences, p 618-630) of the discussion on Noglers hypothesis of A+- factors. This discussion is loosely related to the data presented in the manuscript and therefore although relevant for the general topic of apomixis in Ranunculus, it is irrelevant for the scope of the present manuscript. I would suggest to remove it or reduce its length in order to make more clear the links to the results and the data presented. For example in line 614 it is stated that “transgressive gene expression effects may be influencing the dominance of A ̄” but there is no data, no result and no experimental design enforced to address such claims. The following paragraph (p 618-630) again extensively describes apomictic factors relation to the timing of AI induction, with the last two sentences mentioning transgressive expression without any link to the previous sentences, but more importantly without any link to the new results and new data presented in this manuscript.
  • 640-Description of interesting candidate genes should be in results. Why only these three (EDA, GEX3, and AGO) genes that fell into the transgressive category are the most interesting? Why not other genes with parent of origin effects, or most importantly the genes that were found differentially expressed between sexuals and apomicts in the statistically sound three to three sexual vs. apomict comparison? Please dedicate some lines to discuss your results which are in fact, these list of genes.

Author Response

Reviewer 2: A main concern is the apparent lack of proper experimental design for a transcriptomic study that aimed to identify patterns of differential gene expression. It is widely accepted in the community the use of at least 3 biological replicates per genotype for proper statistical analyses to be performed. This is not the case in this study. Table 1 discloses the samples used and clearly there are 7 individuals that were sampled for 4 stages for a total of 28 samples. The only sample that would have the proper replicates in this case is REV1. Now the authors argue in l. 364 that the comparison of apomictic hybrids and the diploid sexual with both groups having three genotypes each for statistical comparison is valid. This may be accepted but of course the reality is that the hexaploidy apomicts are actually two genotypes and one is not replicated. This biological variance may affect the accuracy of results.

We partially agree on this point. To our understanding the use of biological replicates refers to parallel measurements of biologically distinct samples (i.e. distinct genotypes) that capture random biological variation. The case of taking three samples from the same plant/genotype (or analysing the same sample multiple times) is referred as technical replicates. In our study, we used three biological replicates from three different individuals coming from two populations instead of one like is the case for the diploids.

Even though we used hexaploid genotypes from two populations instead of one, we have to point out that the chosen genotypes are genetically closely related (see Paun et al. 2006, Mol. Ecol. 15) and show the same developmental pattern in apomictic ovules (i.e. they are aposporous and show similar expression in time and space; see Hojsgaard et al. 2014, New Phytologist 204). So, the use of genotypes or biological replicates from two populations was aimed at including apomictic individuals with slightly different proportions of apomixis (see Table 1) to gain on genetic information connected to the expression the trait. This is part of the goal of using distinct samples, to capture random biological variation. On the other hand, the obtained results are not disparate compared to similar studies in other diploid sexual – polyploid apomictic systems.

Regarding the tetraploid cytotype, we might want to emphasize that we would have wanted to have more tetraploid samples included. However, back then we did not have other tetraploid plants with good amounts of flower buds at similar developmental stages. Since the species used in the study are wild species, the reviewer may want to know that the tetraploid is a rather rare cytotype in nature which grows about 800 kms away from our working places, and hence we did not have the chance to collect more materials. Since the Ranunculus plant materials used in this study bloom -more or less- synchronously once per year, having all cytotypes in similar quantitative and qualitative levels at one time window was particularly difficult. Moreover, shifting one year the experiments (till the next flowering season) was not an option for different reasons (including project´s reports and PhD program schedules). Therefore, we decided anyway to collect materials from this single tetraploid genotype aiming at getting at least some extra information about the “sexual” group. This is also the main reason why our analysis of diploid-tetraploid plants is presented with critical observations and without making strong conclusions. We think this is better than not presenting this dataset at all. Our main observations are based on diploid-hexaploid datasets.

Reviewer 2: “Added to the lack of proper replication is the fact that differences in expression may be due to the fact higher ploidy level when comparing the hexaploid apomicts to diploid sexuals…rather than anything to do with apomixis as such. This also can be accepted as polyploidy and apomixis are related and… it is impossible to disentangle its different effects on the gene expression patterns. Care should be taken with conclusion statements and address these uncertainties underlying the grounds for differences in expression.”

We do understand we cannot disentangle effects of apomixis vs. polyploidy vs. hybridity, simply because apomixis is linked to both in our plant system (as in many other systems). So, having each condition separated in materials with sexual and apomictic phenotypes is not a choice. We aimed at collecting evidence on gene expression consequences of individual effects i.e. hybridity or polyploidy, but of course, is it not possible, sensu stricto, to separate these effects in our system. As the reviewer also recognize it in the following commentary, we also think we have addressed the discussion with caution regarding this point, and we have now reformulated several paragraph of the discussion for clarity.

Reviewer 2: “The authors concede the lack of statistical power they have to draw any significant inferences on the role of the tetraploid parental R. cassubicifolius YBB 1. Indeed, any conclusion from this analyses is highly doubtful as there was a SINGLE individual sampled. No replicates at all. I would recommend to refrain to use this sample at all.”

Yes, we do agree. In fact, we made clear that before as the main reason why we did not carry out detailed analysis using the tetraploid or sample (lines 246-251). Simultaneously, we did not include strong statements in the discussion section about the results coming from the inclusion of the tetraploid sample in some analysis. Certainly, we could take the tetraploid sample out of the study. However, since 1) the main results are driven by the diploid-hexaploid analyses, 2) the tetraploid was used in specific cases only to have a hint on possible expression effects, 3) some extra information was collected from including the tetraploid in the analysis (one tetraploid gene with 100% similarity to the hexaploid but not the diploids, lines 531-540), and 4) we did not make strong statements out of it, we would prefer to keep the tetraploid sample in the present study. In our plant system, we think having one sample of the tetraploid putative parent is better than having no sample at all, and as mentioned, it provides few extra information on general comparison between sexuals and apomicts. We do have, however, removed the data of the tetraploid from the Table 3 and included it in a new supporting table (Table S3).

Reviewer 2: “…the methodology implemented for the “expression effects” relies heavily on comparisons and filtering of datasets to expression patterns on this single tetraploid sample, for example as described inlines 247, 252, 258. This type of analyses is tipically done by using the hybrid expression estimates (in this case the hexaploid apomict) and comparing it to the parentals, here the diploid and tetraploid sexuals. In the current analyses filtering and sorting is done by using the tetraploid parent and this seems counterintuitive for me. Please let me know what am I getting wrong here. This needs clarity for further readers and the issue of the lack of replicates of the tetraploid parent needs to be addressed.”

Thank you for this observation. We realized the explanation of the methodology was somehow misleading. We have now rephrased the section (please see lines 252-282) making clear that the main comparison in the “expression effects” analyses is done between sexual diploids and apomict hexaploids. The tetraploid is used in all subsections as an extra comparison. Under “ploidy”, since the single tetraploid have no mean values, we used mean values of the sexual diploid as proxy.

Regarding the lack of tetraploid replicates, we have now rephrased the original sentence in lines 250-251 to: “…due to sample size and lack of biological replicate, a similar statistical comparison (to that of 2x-6x) using the tetraploid sample was not possible”.

Reviewer 2: Formatting of figures is an issue.

Figures 1 and 3 are reformatted for clarity (see more details below).

Reviewer 2: “An important contribution of studies as this is the identity of the transcripts identified in the analysis of differential gene expression and here, those of “expression effects”. These results summarized in table 3 and some supplementary tables are far from informative and most importantly lack the information that would allow researchers to pinpoint these genes in Ranunculus individuals for future studies. It is imperative that these transcripts are properly identified and made traceable for the scientific community.”

We partially agree with the reviewer in this point. We reckon that Table 3 was not clear enough and thus we have now revised the Table and the presentation of results (please check changes in Table 3 and the Result section). However, the aim of Table 3 is to present the data in a more descriptive way to facilitate results interpretation and analysis. More details cannot be provided without increasing the complexity of the table. On the other hand, Tables S1 and S3 contain all information required to facilitate gene identifications. Particularly, for those differentially expressed genes that were annotated, we have presented all information by sequence in Tables S1 (for the four developmental stages = “I, II, III, IV”) and S3 (for developmental stage and expression effect = “grp”), including data on Seq. names, Seq. description and length, Go terms, Enzyme codes, and different statistics. To our understanding, this information should be more than enough for traceability (either back to the transcriptome assembly or in databases).

Reviewer 2: Also, regarding the suggestion on improving annotation reporting and results in general. On the very last part of the discussion the identity of three “choosen” genes is disclosed and discussed. Part of this should be move to results and I suggest to expand on interesting gene candidates that resulted from all analyses.

We have added more details on the differentially expressed genes (please see lines 420-427).

Responses to detailed comments:

Rev2: L 170: The authors declare multiple technical replicates for each gene, but these are all probes that bind to the same original cDNA molecule provided. These are therefore different readings of the same molecule, I would therefore not consider this a technical replicate. There is a single molecule that was

To our understanding, it is not exactly the same molecule. Since the diploid has two alleles, the tetraploid has four, and the hexaploid has 6 alleles, the original cDNA molecule in each case is likely not one.

Rev2: L 181. Sup. Figure 1 does not show any details of the collection stages.

Yes. Thank you. The figure was mislabelled. We have now corrected it.

Rev2: L 205 - figure s3 shows results, remove or clarify

It was mislabelled. It is now Figure S1.

Rev2: L 219 you performed paired t test between sexual and apomicts grouping different genotypes and ploidies in those groups. Each of those samples is an independent genotype that should not be grouped with other samples. I think this analyses are statistically faulty. One wants to dissect the effects of apomixis but here ploidy as well as genotype will affect expression estimates and this is not accounted for. More appropriated analyses ie. Nested designs including genotype and ploidy as factors, could be implemented.

The grouping was between (diploid) sexuals and (hexaploid) apomict, there were no differences in ploidy within groups. Since genotypes within each group are 1) genetically related, 2) they show no phenotypic differences regarding developmental staging and reproductive patterns, and 3) statistical comparison was done for genes expressed within each reproductive stage (meaning we have two sets of observations in each case, i.e. diploid vs. hexaploid), we do think our analysis is appropriate.

We could, perhaps, use a hierarchical analysis in the developmental progression i.e. between samples I, II, III, and IV. This would be rational if we think that genes in stage I will trigger other specific genes in stage II, and so on, having always similar expression levels between individual paired factors. However, in our case, we cannot know a priori what the expression levels will be, and we will lose information on genes differentially expressed but that are crossed rather than nested among stages.

Rev2: 2.4 is difficult to follow. In particular line 227. I understand a gene set is selected based on DGE results. These DEG are selected based on a per stage criteria, as explain in 2.2.3. and then “the expression over all other stages was added to the data set”. Please clarify.

We agree. The paragraph was rephrased (see lines 232-236).

Rev2: On results it is then clear the STEM analyses are made on apomicts and sexuals separately. This should be described the methods section.

This is done on genes expressed differentially between sexual and apomictic ovules stages.

Rev2: L 231 please shortly explain what are the statistics you use in your analyses, the citation of Rapp et al is vague

This is true, but it is also true that Rapp et al. (2009) analysis is complex and include a formula for fitting expression levels to a linear model. We have rephrased the sentence trying to clarify the strategy (see lines 252-257), but if the reviewer considers necessary to provide more specific details we will do so.

Rev2: 249 The use of the tetraploid data and its approximation is statistically faulty the authors claimed that by setting their standard deviations limits is “statistically significant” and I do not think it is, as no test has been performed for comparisons involving the tetraploid samples.  

The paragraph was rephrased (see lines 272-279). Standard deviations for diploids and hexaploids are used to compare mean values against the values of the tetraploid.

Rev2: 250 please shortly elaborate on the classifications described in the reference Yoo et al.

We have now clarified expression categories (see lines 263-265).

Rev2: 264  the comparisons between the sexual samples is not statistically explained in methods. Sample size differs (3 to 1 individuals in table one, cannot account for replication). Nowhere is explained how this test for differential expression is performed. If it is just standard deviations limits, and with these differences in sampling I again claim that the conclusions drawn from these analyses are statistically faulty.

The two paragraphs were rephrased, and more details included (please see subsection 2.3, lines 243-287). We do understand the limitations of lack of tetraploid replicates (it is clearly stated here and along the text). Despite the statistical limitations, we think using one tetraploid is better than excluding this cytotype from the analysis (we have provided more specific reasons above)and we refrain of doing strong statements in conclusions.

Rev2: 5 I find it unfortunate that the annotation job was done so poorly. It is very uninformative to see the supplementary tables filled with NA fields. I find that E ≤ 1-5 is a high threshold and I would regardless of e-value report the top hit for all transcript identified, (and all relevant blast results) and highlight those with a high confidence (ie. E ≤ 1-5) match. Even if no hit would be reported, the fasta sequences for this genes identified in the differential gene expression analyses as well as the genes identified with ploidy, parent of origin and hybridity effects. They should be easily recovered from the reference transcriptome produced and these transcript sequences are a very important part of the results that should be made available for the community in general.

Yes, it is. However, the main problem here is that no genomes relative to Ranunculus are annotated. All specific data about annotated DEGs genes is presented in Tables S1 and S3.

Rev2: L 309 –lack of clarity. How do you identify a subset of genes with a PCA analyses on overall sample variation? Please elaborate on this objective.

This is a simple dispersion analysis. A priori we cannot identify a subset of genes. The analysis shows three separated groups that correspond to ploidy. The sentence was rephrased (line 330).

Rev2: L338-344. What is the objective of making these GO representation/assignment analyses? If the representation analyses does not have power and does not make a contribution to the biological question addressed in this study I suggest to remove this.

Even though a representation analysis was not possible, for a number of different genes we have retrieved useful information.

Rev2: Figure 3 is almost illegible. Needs complete reformatting, probably to be splitted?. Significance test needs to be performed between the boxplots distributions. sample size (n) should be disclose above each box. Etc.

Thank you. We have reformatted Figure 3.

Rev2: 497-502- SNP validation of parent of origin effects are not reported in Methods. This should actually be in methods and results and not squeezed in Discussion

Thank you. We have now placed a description of this analysis in Methods (lines 280-282) and Results (lines 397-401). In addition, the paragraph in the discussion was rewritten (see lines 532-540).

Rev2: There is an extensive part (525- 550, all of 4.2.2 part on ploidy differences, p 618-630) of the discussion on Noglers hypothesis of A+- factors. This discussion is loosely related to the data presented in the manuscript and therefore although relevant for the general topic of apomixis in Ranunculus, it is irrelevant for the scope of the present manuscript. I would suggest to remove it or reduce its length in order to make more clear the links to the results and the data presented. For example in line 614 it is stated that “transgressive gene expression effects may be influencing the dominance of A ̄” but there is no data, no result and no experimental design enforced to address such claims. The following paragraph (p 618-630) again extensively describes apomictic factors relation to the timing of AI induction, with the last two sentences mentioning transgressive expression without any link to the previous sentences, but more importantly without any link to the new results and new data presented in this manuscript.

We have now shortened the section and the mentioned paragraphs (see lines 524-612).

Regarding the specific comment about “…in line 614 it is stated that “transgressive gene expression effects may be influencing the dominance of A ̄” but there is no data, no result and no experimental design enforced to address such claims”, we want to mention that we refer to our observation of genes differentially expressed showing a transgressive expression effect between the diploid sexuals and the hexaploid apomicts, and we discuss the possibility that these effects may influence the dominant expression of apomixis in Ranunculus. This is not a statement, it is just a possibility that we must consider while discussing our data and the current knowledge/hypothesis about apomixis in Ranunculus first, and then about apomixis in general. While discussing we made no conclusions, only express possibilities which may allow us to better understand the overall data and help scientists planning future analyses.

Rev2: 640-Description of interesting candidate genes should be in results. Why only these three (EDA, GEX3, and AGO) genes that fell into the transgressive category are the most interesting? Why not other genes with parent of origin effects, or most importantly the genes that were found differentially expressed between sexuals and apomicts in the statistically sound three to three sexual vs. apomict comparison? Please dedicate some lines to discuss your results which are in fact, these list of genes.

Thank you. We have included a description in results (lines 421-428). These three genes are genes from the group differentially expressed between diploid sexuals and hexaploid apomicts, they were the only annotated genes related to ovule development or reproduction. The three felt (by chance) in the transgressive effect class. We have added a short sentence mentioning this and a few other annotated genes.

Round 2

Reviewer 2 Report

I appreciate the recent work of the authors in their manuscript. All the points that needed clarifications were edited. The manuscript has improved.

I still find that there is a pressing issue of lack of data accessibility, critical for a transcriptomic study as this. Table S3 definitively does not provide any information for traceability. Table S1 presents the results of the Blast2GO program. The authors did not make any amendments or improved their blast job/result display, which I continue to find very poorly performed for current standards. With the information of Table S1 one cannot even trace back to a genebank hit sequence as there are no genebank ids in results (result display is not complete). The authors claimed that information on such tables is enough for one to trace back to the original Ranunculus transcripts but I took the effort to try and find the original transcripts in the dryad repository for Pellino et al (2013) MolEcol. For transcripts number 80 in each of the separate sheets of Table S1 (among others), those were: 360969, 353873, 409308 and 555748. I could not recover a single transcript sequence from the assembly on that repository. Maybe am I searching in the wrong assembly?. Please orient me how a reader could obtain this important data.

As long as the scientific community cannot identify these original transcripts in Ranunculus the manuscript lacks a pivotal part of its contribution. This is imperative to be amended. Once a scientist can access the sequences, if someone has the interest can do the proper blast job themselves, so the blasting amendments are desired but not imperative, but access to the sequences/data is imperative for publication. So, my request to the authors is to please provide a means to recover the transcript sequences of the genes identified in their analyses. Either by properly referencing how to trace back to the reference transcriptome or just depositing the sequences of such transcripts in a public repository.

Figure 3 is still in bad shape and as the other reviewer pointed out it is not the best way to display these results. The boxplots are not sharp and outlier points are blurry making it impossible to discern what was their value. The ideal way to publish this data would be to actually include a table with the expression values obtained. I am not asking for this. In view of the lack of expression estimates provided at least a clear plot that summarizes the data properly should be presented. I get the main point of the boxplots but they are not publication quality at the moment.

Best Regards

Author Response

Please, find below our response to reviewer 2 comments (in bold) to the second revision round.

I still find that there is a pressing issue of lack of data accessibility, critical for a transcriptomic study as this. Table S3 definitively does not provide any information for traceability. Table S1 presents the results of the Blast2GO program. The authors did not make any amendments or improved their blast job/result display, which I continue to find very poorly performed for current standards. With the information of Table S1 one cannot even trace back to a genebank hit sequence as there are no genebank ids in results (result display is not complete). The authors claimed that information on such tables is enough for one to trace back to the original Ranunculus transcripts but I took the effort to try and find the original transcripts in the dryad repository for Pellino et al (2013) MolEcol. For transcripts number 80 in each of the separate sheets of Table S1 (among others), those were: 360969, 353873, 409308 and 555748. I could not recover a single transcript sequence from the assembly on that repository. Maybe am I searching in the wrong assembly?. Please orient me how a reader could obtain this important data.

Thank you for checking this. We couldn´t find those transcripts number 80 being mentioned here, but we do find other transcripts missing, we are not sure why. We have now uploaded to GenBank a new full dataset from all genes in Table S3 (now Table S5), including all genes showing differential expression for each of the three pattern effects (i.e. parent of origin, ploidy and hybrid effects).

We got a feedback from GenBank saying the sequences will be accepted but need to provide more information before getting a repository number and link to be included in the paper (page 9 line 367). We are working on that now. Genes/sequences can be traced back following their IDs.

Figure 3 is still in bad shape and as the other reviewer pointed out it is not the best way to display these results. The boxplots are not sharp and outlier points are blurry making it impossible to discern what was their value. The ideal way to publish this data would be to actually include a table with the expression values obtained. I am not asking for this. In view of the lack of expression estimates provided at least a clear plot that summarizes the data properly should be presented. I get the main point of the boxplots but they are not publication quality at the moment.

Yes. This is probably an effect of having the image pasted in the word file. We have now tried to increase the resolution of the image inserted in the file. In addition, we are submitting as Table S4a-d (page 10 line 410) the txt files of the data used to create the box graphics in Figure 3.

Best Regards

Round 3

Reviewer 2 Report

Thanks for your improvements.

It is important to get the Genebank IDs.